# Lights and Shadows on the Therapeutic Use of Antimicrobial Peptides

**DOI:** 10.3390/molecules27144584

**Published:** 2022-07-18

**Authors:** Denise Bellotti, Maurizio Remelli

**Affiliations:** 1Department of Environmental and Prevention Sciences, University of Ferrara, 44121 Ferrara, Italy; blldns@unife.it; 2Faculty of Chemistry, University of Wrocław, 50-383 Wrocław, Poland; 3Department of Chemical, Pharmaceutical and Agricultural Sciences, University of Ferrara, 44121 Ferrara, Italy

**Keywords:** antimicrobial peptides, AMP, host defence peptides, antibiotic resistance, peptidomimetics

## Abstract

The emergence of antimicrobial-resistant infections is still a major concern for public health worldwide. The number of pathogenic microorganisms capable of resisting common therapeutic treatments are constantly increasing, highlighting the need of innovative and more effective drugs. This phenomenon is strictly connected to the rapid metabolism of microorganisms: due to the huge number of mutations that can occur in a relatively short time, a colony can “adapt” to the pharmacological treatment with the evolution of new resistant species. However, the shortage of available antimicrobial drugs in clinical use is also caused by the high costs involved in developing and marketing new drugs without an adequate guarantee of an economic return; therefore, the pharmaceutical companies have reduced their investments in this area. The use of antimicrobial peptides (AMPs) represents a promising strategy for the design of new therapeutic agents. AMPs act as immune defense mediators of the host organism and show a poor ability to induce antimicrobial resistance, coupled with other advantages such as a broad spectrum of activity, not excessive synthetic costs and low toxicity of both the peptide itself and its own metabolites. It is also important to underline that many antimicrobial peptides, due to their inclination to attack cell membranes, have additional biological activities, such as, for example, as anti-cancer drugs. Unfortunately, they usually undergo rapid degradation by proteolytic enzymes and are characterized by poor bioavailability, preventing their extensive clinical use and landing on the pharmaceutical market. This review is focused on the strength and weak points of antimicrobial peptides as therapeutic agents. We give an overview on the AMPs already employed in clinical practice, which are examples of successful strategies aimed at overcoming the main drawbacks of peptide-based drugs. The review deepens the most promising strategies to design modified antimicrobial peptides with higher proteolytic stability with the purpose of giving a comprehensive summary of the commonly employed approaches to evaluate and optimize the peptide potentialities.

## 1. Introduction

The discovery of penicillin at the beginning of the last century has both drastically reduced the severity and mortality of infections and made other therapies, such as surgical operations or therapies that lower the immune defences, safer. Unfortunately, however, in recent decades the phenomenon of antimicrobial resistance (AMR), mainly caused by the indiscriminate use of antimicrobials (both antibiotics and antifungals) even where not strictly required—such as, e.g., in the case of respiratory diseases of viral origin—has become increasingly widespread [1]. In the most developed countries, a major cause of AMR is the common use of broad-spectrum antibiotics in livestock farms to increase food production. However, the problem of AMR is particularly serious in developing countries, where drug-resistant strains of tuberculosis are rampant [2]. Recent studies estimated that the deaths caused by antimicrobial resistant infections could reach 10 million/year in the world by 2050, with a gross domestic product loss and a cost to the world of about 100 trillion dollars [3,4].

The development of drug-resistant microbial strains is a normal consequence of the high rate of reproduction of bacteria and the selection caused by the extensive use of antimicrobials. A warning in this sense had been clearly expressed by Fleming, Chain and Florey on the occasion of the award of the Nobel Prize in 1945. However, the introduction of new classes of antimicrobials, such as tetracyclines in 1950 or macrolides in 1952, allowed the situation to be kept under control for some time. Unfortunately, the development of drug-resistance has not stopped and has led to the production of strains that are also resistant to more than three different classes of antimicrobials (multi-drug-resistance, MDR) [5]. In addition, in recent years there has been a decrease in the development of new antimicrobial agents due to several factors, including: (i) the low cost of each individual dose; (ii) the short treatment period; (iii) the possible development of resistance mechanisms; and (iv) the uncertain permanence on the market. All of these factors significantly reduce the profit margins of pharmaceutical companies that face high costs and long testing times. As an example, we report that in the period 2017–2019 only five new antibiotic therapies have been approved by the EMA (European Medicines Agency) [5].

From the above it is clear that there is a great need for new strategies to fight microorganisms, especially those resistant to traditional therapies. In this context, great attention has been devoted in recent decades to antimicrobial peptides (AMPs) (see Figure 1) [6,7]. They are also known as host defense peptides (HDPs) and are already present in the market in a small, but rapidly growing, percentage [8]. AMPs predominantly consist of protein fragments of a few dozen of amino acids (average length: 33.26 residues [9]) and are produced by both eukaryotic and prokaryotic species as a part of the first defense line of the immune system [10,11]. AMPs typically contain several basic sites, which give them a net positive charge (on average: +3.32 at neutral pH [9]) and a hydrophobic domain.

The antimicrobial peptide database (APD3, https:/aps.unmc.edu (accessed on 16 December 2021) [12]) already contains more than 3000 AMPs [13]. They are characterized by great variability and can be classified in different ways (Figure 2) according to the biological source from which they derive and the corresponding biosynthesis mechanism, their specific activity or biological functions and properties, their structural characteristics or even the presence of long sequences of repeated amino acids [9,14]. As for the source, they can be produced by superior animals, such as mammals or amphibians, but also by insects or microorganisms. As for the activity, the ADP3 database distinguishes 25 different categories: antibacterial, antiviral, antifungal, antiparasitic, anti-HIV, anti-cancer, anti-diabetic, wound healing, and so on. Of note, their possible antiviral activity has given a great impulse to the research during the recent pandemic of COVID-19. In fact, peptides can act against viruses [15] either by inhibiting their attachment to the cell membrane or by destroying the virus capsule or by inhibiting replication [16]. In relation to coronaviruses, the nasal route has recently been suggested as a useful administration route of antiviral peptides [9].

AMPs can also be classified according to their structure, depending on whether they contain α-helices and/or β-sheets, or if they are unstructured or even cyclic [9]; their secondary structure affects their hydrophobicity and, consequently, their interaction with cell membranes. Finally, some AMPs are rich in a specific amino acid such as glycine, proline, histidine, arginine or tryptophan, and this has considerable implications on their antimicrobial activity [17]. In particular, histidine-rich AMPs show a particular affinity for metal ions essential for the growth of microorganisms [18,19], and therefore their antimicrobial action can be linked to the mechanism known as nutritional immunity [20]. The ability to bind metals can also be exploited to fine-tune the activity of AMPs and thus design very specific new therapies [21,22,23,24].

## 2. Strength Points of AMPs

The use of peptides as drugs is particularly attractive, considering that these molecules existed in nature for millions of years without having developed resistance if not in a limited way [25]. In the case of antimicrobial activity, peptides are generally more efficient than traditional antibiotics [26]. The microbicidal action of AMPs is generally rapid [27] and broad-spectrum [28,29,30,31,32] with a low toxicity [33,34,35]. Of particular interest is the action of AMPs against the so-called ESKAPE bacteria that is the group of six main pathogens responsible for infections of the nosocomial type which are very difficult to treat and therefore very dangerous: *Enterococcus faecium, Staphylococcus aureus, Klebsiella pneumoniae, Acinetobacter baumannii, Pseudomonas aeruginosa* and *Enterobacter* species. [36]. These bacteria are normally MDR and this explains their dangerousness. AMPs are often free from side effects and safer even from the point of view of degradation products, which are smaller, easily eliminated peptides or even single amino acids. Normally, AMPs exert their action without interfering with the microbiota, as is generally the case for traditional antibiotics [37]. Because of their high degradability, AMPs have a low tendency to accumulate in tissues [38]. They are also small molecules with low synthetic costs [25], they particularly exploit the potential of solid phase synthesis, and they are characterized by excellent thermal stability and high solubility in water [26].

## 3. Mechanisms of Action

The activity of AMPs mainly derives from their cationic nature [39,40], which allows them to effectively interact with the membranes of bacteria or fungi, which are negatively charged and rich in lipopolysaccharides and lipoteichoic acid. In fact, most AMPs act by destroying membrane integrity [10,25,28]. The main peptide-membrane interaction models predict that an amphiphilic structure is first formed [39,41,42], resulting from the interaction between the peptide hydrophilic domain and the phospholipid charged groups; in addition, the hydrophobic portions of the peptide interact with the lipid double layer. This can happen in different ways, described by means of the following models [9,28,43] (Figure 3). In the “barrel-stave” model, peptide aggregates penetrate the membrane parallel to the phospholipids, thus forming channels that cause a loss of cytoplasm until the membrane collapses. In the “carpet-like” model, the AMPs accumulate on the surface of the membrane, with which they interact through their lipid domains, forming a sort of carpet; once a threshold concentration is reached, the peptides exert a cleansing action that leads to the rupture of the membrane. The third model is known as the “toroidal pore or wormhole” model: AMPs insert perpendicularly and accumulate in the membrane where they bend to form a circular pore of nanometric size. A model involving the formation of aggregates between peptides and phospholipids has also been described, thus allowing the translocation of AMPs across the membrane. Finally, AMPs can destabilize the bacterial membrane by changing its thickness or causing the aggregation and collapse of phospholipid heads; they can also increase the membrane permeability. Nonmembrane-lytic pathways have also been proposed, where the AMP passes through the membrane and attacks a specific target inside the cell [44]. An often-neglected mechanism which can highly affect the AMP selectivity is peptide aggregation [45]. In fact, aggregation can greatly reduce peptide hydrophobicity and its consequent affinity towards neutral membranes such as those of healthy eukaryotic cells.

The mechanisms of action of Pardaxin (1-22), MSI-78 (4-20), DMPC (1-19) and Cecropin B (1-21), very promising AMPs against the most threatening MDR nosocomial bacterial pathogens, have been investigated recently [46]. The study showed that the first two peptides are the most active against the bacteria tested; they act mainly through the membrane damage and destruction. In particular, Paradaxin is able to spontaneously insert itself into the cytoplasmic bacterial membranes, and this ability is the basis of its activity even at very low concentrations.

## 4. AMPs in Clinical Practice

Several antimicrobial peptides are already present in the market of drugs [14,28,47], often used as “last-resort antibiotics” against multi-drug resistant pathogens; some examples are shown in Table 1.

Gramicidins [48] were initially isolated from the soil bacteria *Bacillus brevis*. Gramicidin D is a mixture of several linear peptide isoforms with 15 residues; gramicidin S is a cyclic decapeptide containing 10 amino acids. Both contain D-amino acids. The activity of gramicidin D is mainly directed to the permeabilization of the bacterial membrane, especially of gram-positive bacteria such as *Bacillus subtilis* and *Staphylococcus aureus*. Gramicidin S is active against both gram-positive and gram-negative bacteria and also fungi, destroying their cell content [48]. Gramicidins are often used in mixtures with other antibiotics for topical treatment of infected wounds or in eye disinfection as eye drops [48]. In high doses they are also toxic to human cells and can cause haemolysis [48].

Polymixins are also used clinically; they are cationic lipopeptide antibiotics [49] that were extracted for the first time in 1947 from *Bacillus polymyxa*. They are lipopeptides with a molecular weight of about 1200 Da, containing a cationic cycle and a tail of three amino acid residues to which fatty acids are attached. The structure is amphiphilic, and this gives the polymyxins the ability to insert into the cell membranes and cause their disintegration. Polymyxins are particularly active against gram-negative bacteria, but also have some neurotoxic activity against humans.

Daptomycin (Cubicin^®^) is a cyclic branched lipopeptide, already approved by the FDA in 2003 as a drug against skin infections caused by methicillin-resistant gram-positive bacteria and then approved in 2006 for the treatment of systemic infections. It is a peptide of 13 amino acid residues, including D-alanine and D-serine, and containing decanoic acid bound to its amino-terminal tryptophan residue [22,50].

Some cationic peptides containing a few dozen amino acid residues are instead in the pre-clinical or clinical evaluation phases. These include the following: omiganan, a synthetic peptide (ILRWPWWPWRRK-NH_2_) especially active against fungal infections of different natures [51]; pexiganan [52,53], a synthetic AMP analogous to magainin with broad-spectrum antibacterial action resulting from the destruction of the cell membrane through the toroidal pore mechanism (see above); nisin [52] a natural peptide expressed by *Lactococcus lactis* and used for decades as a food preservative for its antibacterial action against *Listeria monocytogenes* and other gram-positive bacteria; DPK-060, a human kininogen-derived AMP [54] active for disinfection and the treatment of wounds, has also been investigated for the treatment of external otitis (https://clinicaltrials.gov/ct2/show/NCT01447017 (accessed on 16 December 2021)); LL-37, the human cathelicidin [55], which is characterized by antimicrobial and immune-stimulating/-modulating activity, and is effective for healing venous ulcers [56] and has anticancer properties [57]; and PXL01, a lactoferrin-derived peptide effective for post-operative adhesion care [58]. A very promising application field for AMPs is as anticancer drugs; in fact, they are particularly effective in the destruction of the membrane of cancer cells, which are less developed than normal cells due to accelerated metabolism, through the formation of pores or channels [26,59,60,61,62,63,64,65,66].

## 5. Weak Points of AMPs

Despite the great potential of antimicrobial peptides and the large number of studies concerning their applications in different therapeutic fields, only a rather small number of AMPs have entered clinical practice. This is because, alongside numerous strengths, the AMPs have some weaknesses [8,25,38] or, more generally, show poor ADME (absorption, distribution, metabolism, and excretion) properties [28]. First of all, they are characterized by a short lifetime (normally up to 30 min) due to their rapid degradation by serum proteases and intestinal fluids [67,68,69], accompanied by their rapid removal from the circulation by the liver or kidneys (liver/kidney clearance). In fact, all the peptides are remarkably sensitive to proteolytic enzymes, such as exo-peptidase, which remove one or more residues from the N- or C-terminus (e.g., carboxypeptidases or aminopeptidases,) and endo-peptidases, able of cleaving the backbone peptide bonds (e.g., metalloproteinases or serine proteinases).

An additional weak point is their poor bioavailability for oral use (1–2%) [28,70], due to their hydrophilicity and the consequent difficulty of crossing biological membranes such as the cell membrane, intestinal mucosa or blood-brain barrier [38]. As a consequence, the most common applications are aimed at topical use [28,71] or intravenous, intramuscular or subcutaneous administration [38].

The extraction of natural peptides can be expensive, but this problem is often overcome through solid phase peptide synthesis (SPPS) of the same or model peptides [72]. Sometimes, cytotoxicity, especially due to haemolytic activity, has been reported [28,73]. AMPs can be characterised by a lack of specificity [74,75]. Although rarely verified, the development of resistance is also possible if microorganisms are too frequently exposed to AMPs [76]. Finally, possible immunogenic effects cannot be excluded [38].

In order to overcome their short lifetime, several strategies have been attempted, such as the modification of the peptide structure (see below), or the AMPs encapsulation [77], to block the action of the proteases. Moreover, specificity can be improved by combining AMPs with appropriate vectors which target the peptide at the infection site (STAMPs, Specifically Targeted Antimicrobial Peptides) [37,78,79]. 

## 6. How to Increase Enzymatic Stability and Bioavailability

The fragmentation of peptides by proteases can occur both in the blood and within organs such as the kidneys or liver; more than 550 different proteases have been identified in the human body [8]. The peptides can be sensitive to different types of proteolytic enzymes that operate on different points of the sequence, thus producing many degradation products. Protecting a peptide from a specific type of protease may not be enough to significantly lengthen its lifetime [80]. One of the most effective methods to increase the enzymatic stability of a peptide is to modify its structure, according to different strategies (see Table 2) that can be based either on the replacement of individual amino acid residues or changes in the peptide skeleton, including the protection of the amino and carboxyl terminals or cyclization. A second strategy is to use protease inhibitors [80]. Finally, is it possible to act on the formulation, for example by inserting the peptide in a transport system that reduces or prevents contact with the proteases and allows its release in the site of action (see below) [81].

The susceptibility of a peptide to exo-peptidases largely depends on its N-terminal residue [8]. Peptides ending with Ala, Gly, Met, Ser, Thr or Val are generally more stable than peptides ending with Arg, Asp, Leu, Lys or Phe. In addition, peptides that contain long sequences of Glu, Pro, Ser or Thr are degraded more easily [8]. The most used technique for determining the cleavage points is HPLC-MS/MS (high performance liquid chromatography coupled to tandem mass spectrometry). These investigations allow for the designing of targeted modifications in the peptide sequence. In addition, knowledge of the degradation products is useful for studying their properties and their possible biological activity.

The optimization study on the peptide activity normally starts from a model peptide [80] and can be developed according to different procedures, starting, for example, from the determination of the minimum active sequence (MAS) which corresponds to a truncated analogue at the amino or carboxyl terminal [38]. Computational models, e.g., by means of the quantitative structure-activity relationship (QSAR) approach, can also be produced and investigated [82]. It is also possible to simplify the structure by eliminating or replacing one or more sequences of amino acids—identified for example through alanine scanning (Ala-scan) and/or D-scanning (D-scan)—corresponding to sites sensitive to endo-peptidases [83]. Moreover, there are many possible chemical modifications of a peptide [9,84,85] aimed at optimizing its chemical-physical properties while maintaining the biological activity, according to the so-called “peptidomimetic strategy” [70,86,87,88] (see below).

To make the peptide not attackable by an exo-peptidase, one or both of its terminals can be protected by, for example, N-acylation and/or C-amidation, or the formation of N-pyroglutamate or carbohydrate [38]; these modifications often involve changes in biological activity [83]. The peptide terminals can also be modified by PEGylation (binding of PolyEthyleneGlycol, PEG) [89], which also increases the hydrodynamic size of the peptide by decreasing its renal clearance [38]. Moreover, PEGylation can also improve enzymatic stability by exerting steric hindrance against proteases [83]. As an alternative to PEGylation, sialyation can be performed (modification with polysialic acids, naturally occurring and biodegradable polymers of N-acetylneuraminic acid) [83].

An important modification in the peptide structure is cyclization [38], which can be obtained simply by the formation of a peptide bond between the carboxyl and amino terminals and which makes the peptide resistant to exo-peptidases. However, this is not the preferred route of protection because it may have important implications for biological activity [83]. It is worthy of note that any group of the structure may be involved in cyclization, including chain peptide groups and side groups of some amino acids, such as lysine and ornithine, but also serine and cysteine (e.g., with the formation of a lanthionine bridge). With suitable peptide sequences, a disulphide bridge can be created, and it sometimes produces an improvement in antimicrobial activity [90,91]. In general, the cyclic analogues of a linear peptide are not only more stable against proteases, but also more selective towards the target [70]. One of the oldest cyclic peptides with antimicrobial activity is gramicidin S, a powerful cyclodecapeptide discovered almost eighty years ago and still “in the breach” [92,93].

It is also possible to increase the enzymatic stability by replacing an amino acid residue with a non-proteinogenic amino acid [94]. The choice in this field is very wide [70] (Figure 4): (i) D-amino acids, which are normally not recognized by enzymes [95,96,97]; (ii) N-methyl-α-amino acids (characterized by limited conformational freedom and the reduced ability to form both inter- and intra-molecular hydrogen bonds), which have often shown better pharmacological properties than the originals [98]; (iii) proteinogenic amino acid derivatives with a rigid structure, such as L-4,5,6,7-tetrahydro-1H-imidazo [4,5-c]pyridine-6-carboxylic acid (L-spinacine, Spi) or L-1,2,3,4-tetrahydro-isoquinoline-3-carboxylic acid (Tic) [99]; (iv) β-amino acids, which have a significant influence on the secondary structure of the peptide [22] and that can be distinguished in β^2^, β^3^, homologated (with an extra carbon atom) or isomeric (with the same number of carbon atoms—and therefore also the same molecular mass—of the original) [70]; (v) γ-amino acids, in which two carbon atoms separate the amino and carboxyl groups, characterized by a great conformational versatility and stability towards proteases [22,100,101]; (vi) α-substituted (normally alkylated) amino acids, or α,α-disubstituted glycines [102]; (vii) β-substituted α-amino acids, i.e., alkylated to β carbon and thus equipped with an additional asymmetry centre [103]; and (viii) proline analogues, where the formation of α-helices is hindered by the bending induced in the chain [104].

One or more of the peptide bonds that constitute the peptide backbone can be isosterically or isoelectronically modified in various ways [38,105]: (i) the N-alkylation of nitrogen; (ii) the reduction of carbonyl function replaced by a methylene group; (iii) the replacement of the carbonyl oxygen with either a sulfur atom (endothiopeptide) or a phosphonamide (–P(O)OH–NH–); and (iv) the replacement of the NH group to form a depsipeptide (–CO–O–), a thioester (–CO–S–) or a ketomethylene (–CO–CH_2_–). The whole amide group can be replaced with: (i) a retro-inverso bond (which is equivalent to introducing a D-amino acid, while maintaining the topology of the side chains); (ii) a methylene or thiomethylene bond; (iii) a –CH_2_–CH_2_– bond; or (iv) a hydroxyethylene bond [38]. Furthermore, the α-CH group can be replaced by a nitrogen (azapeptide), which induces a β-turn in the secondary structure [22,70]. An interesting class of modified peptides (or pseudopetides) is that of peptoids, poly-N-substituted glycines where the side chains are not linked to the α-carbons but to the nitrogen atom of the peptide backbone [22,70,106]. This change, in addition to conferring great stability towards proteolytic enzymes, involves the loss of chirality and the absence of the amide hydrogen, often responsible for the formation of hydrogen bonds that determine the secondary structure of the peptide [22]. Modifications to the peptide skeleton include the use of α,β-dehydroamino acids to form dehydropeptides, in which the hydrogen normally present on the carbon of the peptide skeleton has been replaced by a double bond that joins the carbon to the side chain [22].

The bioavailability of an AMP can be improved through a labile modification of its structure, through the so-called prodrug approach [70]. For example, the peptide can be conjugated with a polymer [107] that improves its pharmacokinetics and acts as a shield against enzymes. The additional element can then be removed, for example through proteolytic cleavage, near the active site. Other possible strategies to increase the bioavailability of a peptide, without modifying its structure, are the enzyme inhibition approach and the sustained delivery systems. The former consists of administering the peptide together with specific enzyme inhibitors, in order to increase, for example, absorption by the oral route [83]. In the second case, specific drug carriers are used, such as liposomes [108], ethosomes, transferosomes, cubosomes, nanostructured lipid carriers or solid lipid nanoparticles [81,109]. Alternatively, biodegradable polymers can be used, such as poly(lactic acid) (PLA) or poly(lactic=glycolic acid) (PLGA) [110], emulsions [81] or cyclodextrins derivatives [83].

## 7. Enzymatic Stability and Methods for Its Determination

The susceptibility of a peptide to enzymatic degradation depends on both the amino acid sequence and other factors, such as size, conformation, flexibility, the presence of side chains that can be oxidized or reduced, lipophilicity, and the ability to bind to proteins or other carriers. In the case of parenteral administration, the peptide encounters the proteases present in the blood: the human plasma contains more than one hundred different types of proteins, which also include numerous proteolytic enzymes, classified as esterases and peptidases, which are responsible for the degradation of peptides [38]. In case of oral administration, once past the very acidic environment of the stomach, the peptide enters the lumen of the small intestine which contains large amounts of peptidases secreted by the pancreas. The epithelial cell membrane also constitutes an enzymatic barrier for peptides, as it contains several different types of peptidases [111]. Moreover, if the target of the peptide is the central nervous system, the barrier to be overcome is the blood-brain barrier (BBB) which also contains many enzymes, such as the gamma-glutamyl transpeptidase [107]. Many other tissues and organs, such as the liver and kidneys, also contain enzymes [38,70]. The metabolites produced by peptide proteolysis undergo rapid excretion by the kidneys and liver [28].

The classic method for determining enzymatic stability is to incubate the peptide in 1 mL of human plasma or serum and 1 mL of buffer at pH 7.4, at a temperature of 37 °C [83,112,113]. Aliquots of the mixture are taken at regular intervals and the enzymatic reaction is blocked by the addition, for example, of a strong acid. The sample is then centrifuged, and the supernatant analysed by HPLC-MS. The area of the peak corresponding to the unchanged peptide is monitored and plotted as a function of the incubation time. Of course, this method is useful for determining the degradation in the systemic circulation. To evaluate the action of enzymes present in other organs or tissues, it is possible to incubate peptides in different matrices, such as the liver or kidney homogenates [114], gastrointestinal fluids, intestine or kidney brush border membranes vesicles, liver or kidney microsomes and other more [8]. More detailed information can be obtained by adding individual enzymes to the buffer if they are available on the market or can be extracted and purified [115]. Alternatively, for information on specific enzymes, it is possible to (1) measure the overall effect of proteolysis, for example in plasma, and then (2) to add specific enzyme inhibitors to the reaction medium that block the considered enzyme so as to observe if there are changes in the behavior of the system [83]. The study of enzymatic degradation, in addition to being useful to determine the half-life of the peptide, can also provide information on the fragments generated by degradation, which, in turn, may possess biological activity. Such fragments are usually identified by HPLC-MS; this also allows for the designing of peptide modifications that make it less susceptible to the attack of proteases [116].

The cleavage positions of a protein or peptide can also be investigated by using bioinformatic resources, such as, for example, the PeptideCutter online tool from the ExPASy server (https://www.expasy.org/ (accessed 16 December 2021)). Using this tool to predict the trypsin enzymatic products, Ma and coworkers [117] have been able to propose a new AMP (GV21), corresponding to a truncated sequence of a longer peptide (GV30), originally extracted from the skin secretion of a frog, with lower toxicity and an activity against methicillin-resistant *Staphylococcus aureus* similar to but faster than that of the parent peptide.

It should be borne in mind that all in vitro tests may give results that are over- or under-estimated compared to what happens in vivo [83]. A decisive proof of the real enzymatic stability can only derive from in vivo studies that also give accounts of variations of the lifetime due to modifications of the molecular mass or the hydrophobicity, asit happens with PEGylation. However, it is worthy of note that new in silico tools for converting the results of in vitro toxicity tests into reliable predictions of in vivo behaviour are rapidly growing [118], although their full application on a large scale is currently limited by the incomplete knowledge of all the metabolic mechanisms involved in defining the toxicity of a drug.

## 8. Stability and Half-Life of AMPs: A Case Study

In the presence of proteases, the half-life of peptides is generally of the order of minutes, only rarely of a few hours [38,99]. Lifetimes can greatly increase as a result of changes in the peptide amino acid sequence with the introduction, for example, of a non-proteinogenic amino acid or as a consequence of the protection of its termini [119]. In this regard, a very interesting systematic study [120] examined the effect of point modifications on the enzymatic stability of a selected peptide. The modification consisted in the introduction of a non-proteinogenic residue, i.e., either a D-amino acid, a methylated amino acid or a β^3^ residue. A library of 32 mutants of the original peptide was synthesized, varying the position and type of introduced modification and performing tandem substitutions. All mutants were then studied under the same experimental conditions, by incubation with chymotrypsin, as representative of the vast family of serine proteases, a class of enzymes among the most abundant in the human body. The original peptide sequence was chosen by excluding amino acids with redox activity, such as Cys, or particularly hydrophobic, such as Met or Phe, and introducing instead a cleavage site typical of chymotrypsin (AYK) at the centre of a sequence of a total of 21 amino acids (VSADPSRVNAYKSADSRVNST). This peptide is divided into two fragments by the enzyme, by breaking the peptide bond between Tyr11 and Lys12, with a lifetime of 8 min under the selected experimental conditions. No breakage was observed in other parts of the peptide. The substitution of Tyr11 with a non-proteinogenic amino acid increased the lifetime by 100–1000 times; less obvious was the effect of the substitution of Lys12. The introduction of an N-Me-α residue was generally the least effective modification, while the protection action due to the presence of a C_α_-methylated amino acid or a β^3^ residue was better and acted over a longer range. However, the most effective modification was the introduction of a D-amino acid, even in locations far from the cleavage site. The study of disubstituted analogues in different positions suggested the possibility of a combined synergistic effect, in either a positive or a negative sense, that was not easily predictable. Overall, this study showed that the protective action due to the substitution of an amino acid with a non-proteinogenic residue may be of either a local type, namely a specific interference in the enzyme-substrate interaction, or a global type, which involves the entire structure of the denatured peptide. While completely replaced peptides, such as D-α-peptides or β-peptides, are inert to protease, the stability of only partially replaced peptides is in some way related to the extent of such substitution as well as to the type of substituent used. Unfortunately, in general, the biological activity decreases as the structure of the peptide changes from the original one.

## 9. Conclusions

Antimicrobial peptides can be a very effective answer to the highly serious problem of resistance to antimicrobial drugs: they are very promising candidates for the design of novel classes of therapeutic agents, mostly due to their scarce attitude to induce resistance in pathogenic microorganisms. The enormous variability with which the peptide sequence can be designed and modified extends in an almost unlimited way the application field of AMPs. In fact, with only small structural alterations, they can be made resistant to proteases and their activity can be modulated, not only against micro-organisms but also for the treatment of other diseases. In addition, several AMPs have the ability to chelate metal ions, and this extends their possible mechanisms of activity to, for example, that of nutritional immunity. This research field is still largely unexplored, and a great advancement can be expected in the coming decades.

## Figures and Tables

**Figure 1 molecules-27-04584-f001:**
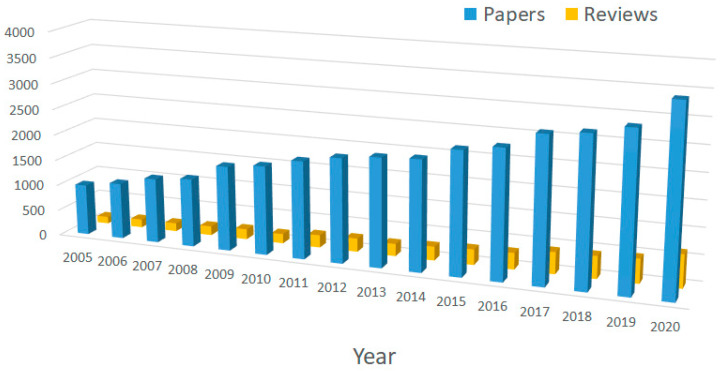
Published research paper reported in the Web of Science (https://www.webofscience.com/wos/woscc/basic-search (accessed on 16 December 2021)) between the years 2005 and 2020, containing the words “antimicrobial peptide” in the topic.

**Figure 2 molecules-27-04584-f002:**
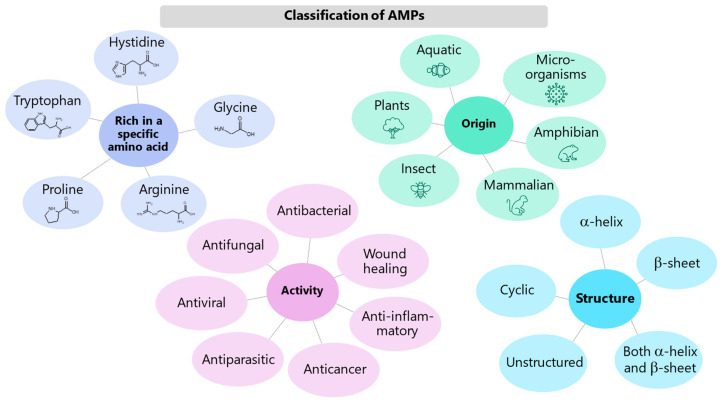
Classification of AMPs.

**Figure 3 molecules-27-04584-f003:**
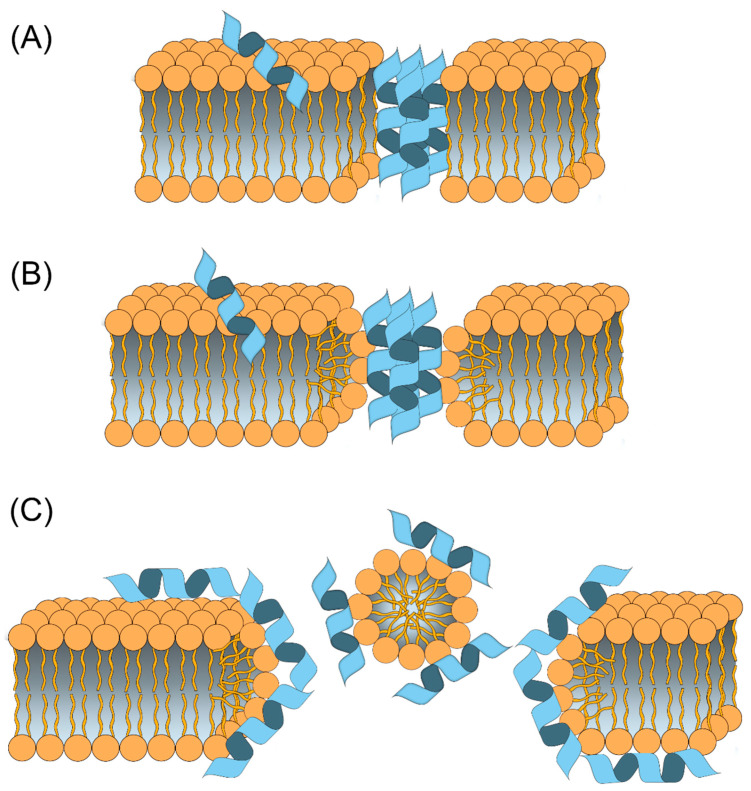
Most widely accepted models for extracellular AMPs mechanisms of action. (**A**) Barrel-stave model: AMPs aggregate as multimers and arrange parallel to the phospholipids of the cell membrane to form a hydrophilic transmembrane channel. (**B**) Toroidal pore model: AMPs accumulate vertically within the membrane causing the lipid moieties to fold inward to form a circular pore. (**C**) Carpet model: AMPs interact with the surface and act as a “detergent” isolating different portions of the membrane.

**Figure 4 molecules-27-04584-f004:**
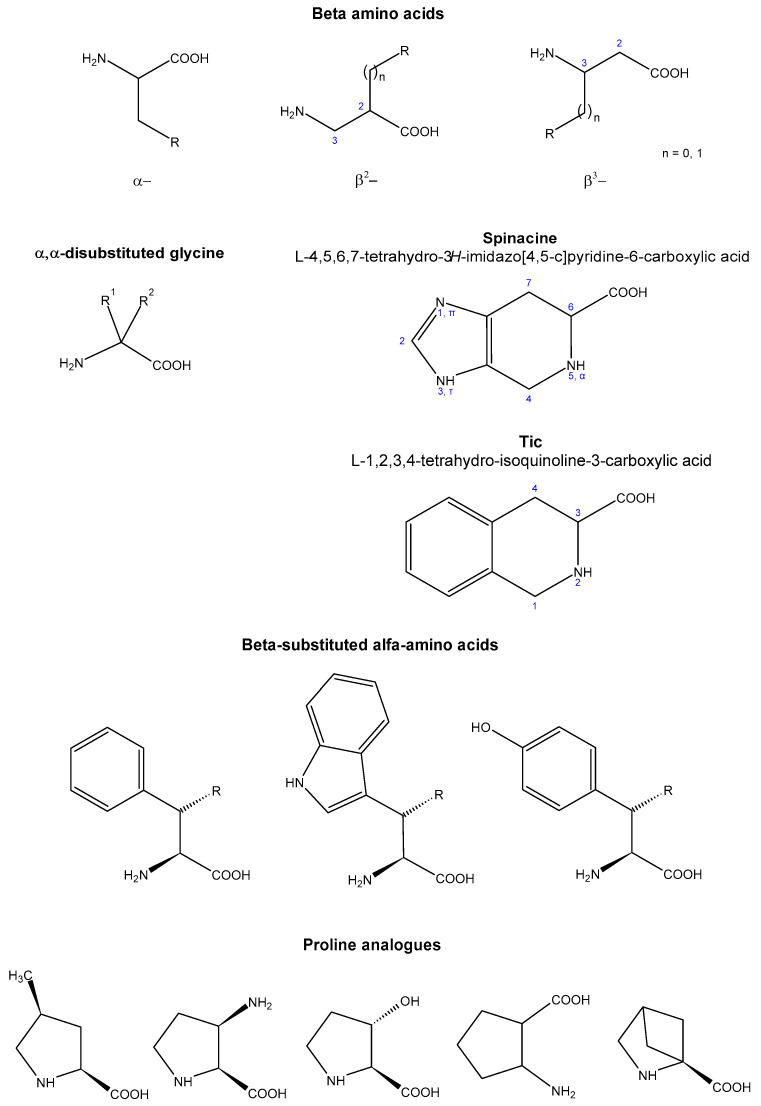
Some non-proteinogenic amino acids.

**Table 1 molecules-27-04584-t001:** Examples of AMPs in clinical practice and/or clinical and pre-clinical evaluation.

Name	Structure	Antimicrobial Activity	Ref
Approved AMPs			
Gramicidin D	Linear peptide isoforms with 15 residues (VGLAVVVWLWLWLWG)	Gram-positive bacteria	[48]
Gramicidin S	Cyclic decapeptide (cyclo[LFPVOrnLFPVOrn])	Gram-positive and gram-negative bacteria, fungi	[48]
Polymyxins	Lipopeptides with a cationic cycle and a tripeptide chain N-acylated by a fatty acid tail	Gram-negative bacteria	[49]
Daptomycin	Cyclic branched 13-mer lipopeptide	(Methicillin-resistant-) Gram-positive bacteria	[22,50]
AMPs in clinical and pre-clinical evaluation			
Omiganan	Linear 12-mer cationic peptide (ILRWPWWPWRRK-NH_2_)	Fungi	[51]
Pexiganan (MSI-78)	Linear 22-mer cationic peptide (GIGKFLKKAKKFGKAFVKILKK-NH_2_)	Gram-positive and gram-negative bacteria, fungi	[52,53]
Nisin (e.g., Nisin A)	Linear 34-mer cationic peptide (ITSISLCTPGCKTGALMGCNMKTATCHCSIHVSK)	Gram-positive bacteria	[52]
DPK-060	Linear 20-mer cationic peptide (GKHKNKGKKNGKHNGWKWWW)	Gram-positive and gram-negative bacteria	[54]
Human cathelicidin (LL-37)	Linear 37-mer peptide (LLGDFFRKSKEKIGKEFKRIVQRIKDFLRNLVPRTES)	Gram-positive and gram-negative bacteria, fungi	[55,56,57]
PXL01	Lactoferrin-derived peptide formulated in sodium hyaluronate	Adhesion inhibition	[58]

**Table 2 molecules-27-04584-t002:** Principal strategies to enhance the proteolytic stability of AMPs.

**Terminal modification**	N-acylation, C-amidation, formation of N-pyroglutamate and carbohydrate, PEGylation, sialyation.
**Cyclization**	Head-to-tail cyclization, head-to-side-chain cyclization, side-chain-to-side-chain cyclization (e.g., disulphide and lanthionine bridge formation).
**Replacement of one or more residues with non-proteinogenic amino acid**	D-amino acids, N-methyl-α-amino acids, proteinogenic amino acid derivatives with a rigid structure (e.g., Spi, Tic), β-amino acids, γ-amino acids, α-substituted amino acids, β-substituted α-amino acids, proline analogues.
**Formation of pseudopetides** **(replacing peptide bonds with other chemical groups)**	N-alkylation, carbonyl function substitution with a methylene group, carbonyl-O substitution with a sulfur atom or phosphonamide, NH group substitution with oxygen (depsipeptide), sulfur (thioester) or methylen (ketomethylene). Introduction of a retro-inverso peptide bond, methylene or thiomethylene bond, a –CH_2_–CH_2_– bond, or a hydroxyethylene bond. Formation of azapeptides, peptoids and dehydropeptides.
**Elimination of one or more residues**	Deletion of amino acid residues which are more susceptible to proteolytic attack.
**Prodrug approach**	Introduction of a labile modification, maintaining the peptide structure almost unchanged by means of peptide conjugation with a polymer.
**Use of protease inhibitors**	Co-administration of the peptide and a specific enzyme inhibitor.
**Formulation modification**	Application of specific drug carriers (liposomes, ethosomes, transferosomes, cubosomes, nanostructured lipid carriers, solid lipid nanoparticles, biopolymers).

## Data Availability

Not applicable.

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
