# Peer review of "Lights and Shadows on the Therapeutic Use of Antimicrobial Peptides"

_molecules, 2022, doi:10.3390/molecules27144584_

Round 1

Reviewer 1 Report

This review about AMPs brings nothing new. They are already several recent reviews on the same topics (for example, https://doi.org/10.3389/fmicb.2021.616979, https://doi.org/10.1093/database/baac011, https://doi.org/10.3389/fmicb.2020.582779, https://doi.org/10.3390/antibiotics11030349, https://doi.org/10.1099/mic.0.001136). The authors should emphasize what is really new in their review. I think they go after the surface, they do not in into detail on any topic. I would pick one topic, for example, the pharmacokinetics of AMPs (which has been not elaborated on in detail so far), and systematically review related knowledge. Without that, I do not recommend publishing this review.

1. The authors write “Recent studies estimated that the deaths caused by antimicrobial resistant infections could reach 10 million/year in the world around the year 2050, with a gross domestic product loss of more than 100 trillion dollars” (L47-L50). But, in the cited literature, these sentences are stated “The potential costs to human life and to the economy are sobering. Recent U.K. government–commissioned reports (6, 7) estimate that if no action is taken, by 2050, AMR will cause up to 10 million annual deaths globally, reduce gross domestic product (GDP) by 2 to 3.5%, and cost US$100 trillion.” or “The results show a considerable human and economic cost. Initial research, looking only at part of the impact of AMR, shows that a continued rise in resistance by 2050 would lead to 10 million people dying every year and a reduction of 2% to 3.5% in Gross Domestic Product (GDP). It would cost the world up to 100 trillion USD.” That means GDP will reduce by 2 to 3.5%, not more than 100 trillion dollars. The authors should rephrase the literature correctly.

2. The sentence “a research field of great growth in recent decades (see Figure 1) is that of antimicrobial peptides (AMPs)” (L67-L69) should be rephrased.

3. In two sentences about Gramidicin D (L156-L158), references are missing.

4. It is not clear based on what the authors select examples of AMPs in clinical practice.  I suggest making a table with them. Additionally, the section is hard to read.

5. The authors describe the general activity of some AMPs in clinical practice. But they are being studied for the specific application which the authors omit. For example, DPK- 060 (L185-186) is being specifically investigated for its clinical safety and efficacy in patients with acute external otitis, not for disinfection and treatment of wounds. The authors should rewrite this section.

6. The authors stated in one place, that “In the case of antimicrobial activity, peptides are often more efficient than traditional antibiotics” and in the other one, that “AMPs can be characterised by a lack of specificity [72,73], accompanied by a lower antimicrobial activity than traditional antibiotics.” (L215-L216). They should explain how they mean this discrepancy and synchronized these sentences.

7. I am missing in silico modeling of enzymatic stability in the methods for its determination.

8. The authors suggested animals as the best model for in vivo experiments omitting problems that animals are not humans and the extrapolation to humans is critical (L373). Rapid progress in the development of new in silico and in vitro methods facilitates the movement away from animal studies even in pharmacokinetics (e.g. https://doi.org/10.3389/ftox.2022.894569). The authors should address that.

9. The whole chapter “Stability and half-life of AMPs” is based on one study. I am not sure why this study is so important. I would include more studies focusing on the structural features of AMPs and half-life and biological activities combine this chapter with the previous one.

Author Response

This review about AMPs brings nothing new. They are already several recent reviews on the same topics (for example, https://doi.org/10.3389/fmicb.2021.616979, https://doi.org/10.1093/database/baac011, https://doi.org/10.3389/fmicb.2020.582779, https://doi.org/10.3390/antibiotics11030349, https://doi.org/10.1099/mic.0.001136). The authors should emphasize what is really new in their review. I think they go after the surface, they do not in into detail on any topic. I would pick one topic, for example, the pharmacokinetics of AMPs (which has been not elaborated on in detail so far), and systematically review related knowledge. Without that, I do not recommend publishing this review.

We thank the Reviewer for the suggestion of new references to add to the paper. He/she is absolutely right that many reviews on the same or a similar topic are published almost every day, and this is a proof that the topic is very hot. The “pharmacokinetics of AMPs” is a very good suggestion for a new review. The aim of the present review is to give an overview of the last literature on AMPs with an especial attention to the problem of enzymatic stability and the “tricks” to overcome it.

  1. The authors write “Recent studies estimated that the deaths caused by antimicrobial resistant infections could reach 10 million/year in the world around the year 2050, with a gross domestic product loss of more than 100 trillion dollars” (L47-L50). But, in the cited literature, these sentences are stated “The potential costs to human life and to the economy are sobering. Recent U.K. government–commissioned reports (6, 7) estimate that if no action is taken, by 2050, AMR will cause up to 10 million annual deaths globally, reduce gross domestic product (GDP) by 2 to 3.5%, and cost US$100 trillion.” or “The results show a considerable human and economic cost. Initial research, looking only at part of the impact of AMR, shows that a continued rise in resistance by 2050 would lead to 10 million people dying every year and a reduction of 2% to 3.5% in Gross Domestic Product (GDP). It would cost the world up to 100 trillion USD.” That means GDP will reduce by 2 to 3.5%, not more than 100 trillion dollars. The authors should rephrase the literature correctly.

We agree that the sentence was misleading, and it has been changed to “Recent studies estimated that the deaths caused by antimicrobial resistant infections could reach 10 million/year in the world around the year 2050, with a gross domestic product loss and a cost for the world of about 100 trillion dollars”

2. The sentence “a research field of great growth in recent decades (see Figure 1) is that of antimicrobial peptides (AMPs)” (L67-L69) should be rephrased.

The sentence has been changed to. “great attention has been devoted in recent decades to antimicrobial peptides (AMPs) (see Figure 1)”

3. In two sentences about Gramidicin D (L156-L158), references are missing.

That’s because the reference is always the same; anyway, in the revised paper it has been repeated for the sake of clearness.

4. It is not clear based on what the authors select examples of AMPs in clinical practice. I suggest making a table with them. Additionally, the section is hard to read.

Thank you for this suggestion. A Table has been added.

5. The authors describe the general activity of some AMPs in clinical practice. But they are being studied for the specific application which the authors omit. For example, DPK- 060 (L185-186) is being specifically investigated for its clinical safety and efficacy in patients with acute external otitis, not for disinfection and treatment of wounds. The authors should rewrite this section.

We thank the Reviewer for the suggestions.
Actually, DPK-060 (also known as GKH17-WWW), was originally developed in 2009 for the treatment of skin infections (https://doi.org/10.1074/jbc.M109.011650). In 2011, DPK-060 was investigated for the treatment of external otitis, and the results of this Clinical Trial are available in the literature (https://clinicaltrials.gov/ct2/show/NCT01447017); however, this application have never been cited in the scientific literature (Web of Science, PubMed) afterwards. Instead, the more recent papers report the efficacy of DPK-060 for topical treatment.

The sentence has been modified, accordingly.

6. The authors stated in one place, that “In the case of antimicrobial activity, peptides are often more efficient than traditional antibiotics” and in the other one, that “AMPs can be characterised by a lack of specificity [72,73], accompanied by a lower antimicrobial activity than traditional antibiotics.” (L215-L216). They should explain how they mean this discrepancy and synchronized these sentences.

We thank the Reviewer for this observation. The first sentence has been modified to: “In the case of antimicrobial activity, peptides are generally more efficient than traditional antibiotics”.

At the lines 215-216, the sentence: “, accompanied by a lower antimicrobial activity than traditional antibiotics” has been eliminated.

7. I am missing in silico modeling of enzymatic stability in the methods for its determination.

We thank the Reviewer for suggesting this improvement. After line 375, the following sentence has been inserted: “The cleavage positions of a protein or peptide can also be investigated by using bioinformatic resources, like, e. g., the PeptideCutter online tool from the ExPASy server (https://www.expasy.org/). Using this tool to predict the trypsin enzymatic products, Ma and coworkers [Ma, Yao et al. 2021] have been able to propose a new AMP (GV21), corresponding to a truncated sequence of a longer peptide (GV30), originally extracted from the skin secretion of a frog, with lower toxicity and an activity against methicillin-resistant Staphylococcus aureus similar to but faster than that of the parent peptide.”

8. The authors suggested animals as the best model for in vivo experiments omitting problems that animals are not humans and the extrapolation to humans is critical (L373). Rapid progress in the development of new in silico and in vitro methods facilitates the movement away from animal studies even in pharmacokinetics (e.g. https://doi.org/10.3389/ftox.2022.894569). The authors should address that.

We thank the Reviewer for this suggestion. The end of par. 7 has been modified as follows.
“It should be borne in mind that all in vitro tests may give results that are over- or under-estimated compared to what happens in vivo [81]. A decisive proof of the real enzymatic stability can only derive from in vivo studies, that also give account of variations of the lifetime due to modifications of the molecular mass or the hydrophobicity, like it happens with PEGylation. However, it is worth of note that in silico new tools for converting the results of in vitro toxicity tests into reliable predictions of in vivo behaviour are rapidly growing [Moreau 2022], although their full application on a large scale is currently limited by not complete knowledge of all the metabolic mechanisms involved in defining the toxicity of a drug.”

9. The whole chapter “Stability and half-life of AMPs” is based on one study. I am not sure why this study is so important. I would include more studies focusing on the structural features of AMPs and half-life and biological activities combine this chapter with the previous one.

Actually, it is not easy to found in the literature research paper as detailed as that of Werner et al.. We agree with the Reviewer that the title of the chapter can be misleading, so it has been changed to:
“Stability and half-life of AMPs: a case study”

Reviewer 2 Report

gentlemen authors, this review article fulfilled all my expectations of the therapeutic potential of antimicrobial peptides, as well as their limitations and possibilities of modification to reduce susceptibility to proteolysis such as the use of D-aa, as well as cycling, methylation from the carboxyl or amino ends of amino acid sequences. It is a good article for people who work in this field, as well as for neophytes who are entering this beautiful field of research into new antimicrobial products.

Author Response

We thank very much Reviewer 2 for her/his kind words.

Reviewer 3 Report

The submitted review summarised the advance and limitations of antimicrobial peptide development. It is a very good summary and I really enjoyed reading it. There are only a few minor comments to be addressed.

1.       The text in figure 2 is too small to read

2.       In section 6, it will be better to list a table of the strategy to enhance stability

3.       Few antimicrobial peptide should be AMP, such as line 116,125,322

4.       Line 127, the introduction part of AMPs, a recent detailed investigation on the AMPs mode of actions should be discussed, such as Eur J Med Chem. 2022 Mar 5;231:114135. doi: 10.1016/j.ejmech.2022.114135.

5.       Line 143, a review on aggregation should be referred. Biochimica et Biophysica Acta (BBA) Biomembranes, Volume 1862, Issue 2, 1 February 2020, 183107

Author Response

The submitted review summarised the advance and limitations of antimicrobial peptide development. It is a very good summary and I really enjoyed reading it. There are only a few minor comments to be addressed.
1. The text in figure 2 is too small to read
The font used in the Figure has been expanded.
2. In section 6, it will be better to list a table of the strategy to enhance stability
Table 2 has been added.
3. Few ” antimicrobial peptide” should be “AMP”, such as line 116,125,322
Done, thank you.
4. Line 127, the introduction part of AMPs, a recent detailed investigation on the AMPs mode of actions should be discussed, such as Eur J Med Chem. 2022 Mar 5;231:114135. doi:10.1016/j.ejmech.2022.114135.
We thank the reviewer for the suggestion. The reference has been inserted at the end of the section and commented as follows:
“The mechanisms of action of Pardaxin (1-22), MSI-78 (4-20), DMPC (1-19) and Cecropin B (1-21), very promising AMPs against the most threatening MDR nosocomial bacterial pathogens, have been recently deeply investigated [Lin 2022]. The study showed that the first two peptides are the most active against the bacteria tested; they act mainly through the membrane damage and destruction. In particular, Paradaxin is able to spontaneously insert itself into the cytoplasmic bacterial membranes and this ability is the basis of its activity even at very low concentrations. ”
5. Line 143, a review on aggregation should be referred. Biochimica et Biophysica Acta (BBA) – Biomembranes, Volume 1862, Issue 2, 1 February 2020, 183107
We thank the reviewer for the suggestion. The reference has been inserted at line 146 and commented as follows:
“An often-neglected mechanism which can highly affect the AMP selectivity is peptide aggregation [Vaezi 2020]. In fact, aggregation can greatly reduce peptide hydrophobicity and its consequent affinity towards neutral membranes such as those of healthy eukaryotic cells.”

Round 2

Reviewer 1 Report

The authors addressed most of my comments but they did not explain well what their review brings new. They should add 1-2 sentences about it in the introduction. The authors also do not properly distinguish between clinical applicability of AMPs being assessed in clinical trials and their activities and applicability studied in the basic biomedicine research (Chapter 4). It should be clear where they write about AMP activities from basic research and where about their clinical trials. There are several current more comprehensive reviews summarizing clinical trials for AMPs and their specific clinical applicability (e.g. https://doi.org/10.3389/fmicb.2021.616979 or https://doi.org/10.1002/pep2.24122). They can at least add the references to them.
